# Chronic Low or High Nutrient Intake and Myokine Levels

**DOI:** 10.3390/nu15010153

**Published:** 2022-12-29

**Authors:** Ana Paula Renno Sierra, Antônio Alves Fontes-Junior, Inês Assis Paz, Cesar Augustus Zocoler de Sousa, Leticia Aparecida da Silva Manoel, Duane Cardoso de Menezes, Vinicius Alves Rocha, Hermes Vieira Barbeiro, Heraldo Possolo de Souza, Maria Fernanda Cury-Boaventura

**Affiliations:** 1School of Physical Education and Sport, University of São Paulo, Sao Paulo 05508-030, Brazil; 2Interdisciplinary Post-Graduate Program in Health Sciences, Institute of Physical Activity and Sports Sciences, Cruzeiro do Sul University, Sao Paulo 01506-000, Brazil; 3Emergency Medicine Department, LIM-51, University of São Paulo, Sao Paulo 01246-903, Brazil

**Keywords:** nutrition, exercise, endurance, musculoskeletal, physiology

## Abstract

Inadequate nutrient availability has been demonstrated to be one of the main factors related to endocrine and metabolic dysfunction. We investigated the role of inadequate nutrient intakes in the myokine levels of runners. Sixty-one amateur runners participated in this study. The myokine levels were determined using the Human Magnetic Bead Panel from plasma samples collected before and after the marathon. Dietary intake was determined using a prospective method of three food records. The runners with lower carbohydrate and calcium intakes had higher percentages of fat mass (*p* < 0.01). The runners with a sucrose intake comprising above 10% of their energy intake and an adequate sodium intake had higher levels of BDNF (*p* = 0.027 and *p* = 0.031). After the race and in the recovery period, the runners with adequate carbohydrate intakes (g/kg) (>5 g/kg/day) had higher levels of myostatin and musclin (*p* < 0.05). The runners with less than 45% of carbohydrate of EI had lower levels of IL-15 (*p* = 0.015) and BNDF (*p* = 0.013). The runners with higher cholesterol intakes had lower levels of irisin (*p* = 0.011) and apelin (*p* = 0.020), and those with a low fiber intake had lower levels of irisin (*p* = 0.005) and BDNF (*p* = 0.049). The inadequate intake influenced myokine levels, which promoted cardiometabolic tissue repair and adaptations to exercise.

## 1. Introduction

Exercise promotes the release of chemical messengers as a result of skeletal muscle contraction, called myokines and/or exerkines. The myokines modulate muscle mass, function, and regeneration by acting on protein synthesis, insulin sensitivity, fat oxidation, myogenesis, mitochondrial biogenesis, autophagy, mitophagy, and the remodeling of the extracellular matrix [1,2,3,4].

In addition, many myokines seem to contribute to cardiometabolic adaptations to exercise as a result of crosstalk with adipose tissues, the liver and heart contributing to glucose homeostasis, the browning of white adipose tissue (WAT), and cardioprotection [1,5,6,7].

More than 650 myokines have been described in response to exercise, and many researchers have been investigating their biological effects on different tissues. The most studied myokines include IL-6, IL-15, myostatin, leukemia inhibitory factor (LIF), secreted protein acidic rich in cysteine (SPARC), myonectin, monocyte chemoattractant protein-1 (MCP1), irisin, apelin, decorin, musclin, growth differentiation factor 15 (GDF-15), brain-derived neurotrophic factor (BDNF), fibroblast growth factor (FGF)-21, follistatin (FSTL), meteorin-like (Metrnl), fractalkine, and angiopoietin-like protein 4 [2,3,8]. The myokine levels are dependent on various forms of exercise and training [9].

Nutritional interventions such as caloric restriction or supplementation seem to modulate cytokines, adipokines, myokines, and cardiomiokynes [4,10,11]. Previous studies have reported a poor daily intake with low carbohydrate, dietary fiber source, fruit, dairy beverage, and vegetable intake in endurance runners [12,13,14,15]. Chronic low or higher nutrient availability promotes endocrine and cardiometabolic dysfunction [5,10].

Inadequate daily intake (DI) may influence the myokine response induced by endurance exercise, affecting the dynamics of muscle repair and cardiometabolic adaptations. The maladaptive response after endurance exercise may impair muscle function, performance, and health, or even increase the risk of acute cardiovascular events [16]

The aim of this study to investigate the effects of chronic low or high nutrient intakes on myokine levels before and after endurance exercise.

## 2. Material and Methods

### 2.1. Subjects

Seventy-four amateur Brazilian male marathon finishers (aged 30 to 55 years) participated in this study. The volunteer recruitment was performed by e-mail to all marathon runners registered in the São Paulo International Marathon in 2017 or 2018. Researchers randomly contacted volunteers to confirm their interest and availability to participate in all steps of the study (before the race, immediately after the race, and in the recovery period). Inclusion criteria were training more than 30 km per week and having previously participated in a half marathon or marathon, as well as not having cardiovascular, pulmonary, or kidney injury, and/or liver, kidney, inflammatory, or neoplastic diseases, or use alcohol or drugs.

The Ethics Committee of Cruzeiro do Sul University, Brazil (Permit Number: 3.895.058) approved this study in accordance with the Declaration of Helsinki. All volunteers read and signed the written informed consent document before starting to participate in the study.

Body composition, cardiopulmonary function, and DI were evaluated before the race. Of the seventy-four marathon runners, sixty filled in three food records before the marathon race (one week) for DI analyses. Therefore, we excluded fourteen runners from the analysis of the association between myokines and DI.

The São Paulo International Marathon began at 07:30 a.m. on 9 April 2017 and 8 April 2018. Fluid ingestion was provided during the race (water every 2 to 3 km; sports drinks at 12 km, 21.7 km, 33 km, and 42 km; and a carbohydrate at 28.8 km). The weather during the race was temperate (average temperature and humidity 19.8 °C, 72.8% in 2017, and 19.9 °C, 87.7% in 2018) (National Institute of Meteorology, Ministry of Agriculture, Livestock, and Supply).

An electronic digital scale platform (marte^®^, Sao Paulo, SP, Brazil) was used to measure body mass (kg) and height (cm). Body mass index (BMI) was calculated according to the International Society for Advancement of Kinanthropometry (ISAK) standard [weight (kg)/height (m^2^)]. The body composition was determined by bioimpedance analysis (Biodynamics Corporation, Shoreline (WA), USA, 310e) 24 h before the marathon race in the fasting state.

### 2.2. Cardiopulmonary Exercise Test

After the medical history data collection, the cardiopulmonary exercise test (CPET) was realized between three and one week before the São Paulo International Marathon by a progressive treadmill test with a constant incline of 1%, and an initial speed of 8 km·h^−1^, with an elevation of 1 km·h^−1^ every 1 min until voluntary exhaustion (TEB Apex 200, TEB, São Paulo, Brazil, speed 0–24 km/h, grade 0–35%). The volunteers were monitored with a standard 12-lead computerized electrocardiogram during the test (TEB^®^, ECG São Paulo, Brazil) to rule out any cardiac dysfunction at rest and during exertion. The respiratory gas exchange was measured through open-circuit and automatic, indirect calorimetry (Quark CPET, COSMED^®^, Rome, Italy). The VO2 max of the subjects was determined according to the American College of Sports Medicine [17]. All volunteers finished the International Marathon of São Paulo in 2017 (40 runners) and 2018 (34 runners).

### 2.3. Blood Sampling

Blood samples (10 mL) were collected 24 h before, and 24 h and 72 h after the race from the antecubital vein at the Cruzeiro do Sul University with at least 12 h without physical activity and from fasting runners. To obtain plasma samples, vacuum tubes containing ethylenediaminetetraacetic acid (10 mL, EDTA, 1 mg/mL) samples were immediately centrifuged at 4 °C, 400× *g* for 10 min and then stored at −80 °C for the later analysis of myokines at University of São Paulo. Immediately after the race, blood samples (10 mL) from the fed runners were maintained on ice for approximately 2 h at the International Marathon of São Paulo (competition venue, close to finish line) and then sent to Cruzeiro do Sul University (10 mL) for plasma collection as described above.

### 2.4. Determination of Myokines

Apelin, BDNF, FSTL, FGF-21, IL-6, IL-15, irisin, myostatin, and musclin plasmatic levels were evaluated using the MILLIPLEX^®^ Human Myokine Magnetic Bead (MagPlex^®^-C microspheres) Panel protocol (HCYTOMAG-56K, EMD Millipore Corporation, MA, USA). The fluorescent-coded magnetic beads contain a specific capture antibody of each myokine on the surface with internally color-coded microspheres with two fluorescent dyes detected by Luminex^®^ xMAP^®^ technology. After the capture antibody incubation, a biotinylated detection antibody is added on the assay, followed by a reaction with a Streptavidin-PE conjugate. The high-speed digital signal processors of capture and detection components were analyzed by a Luminex^®^ analyzer (MAGPIX^®^). The intra-assay precision (mean coefficient variation percentage) described by the MILLIPLEX^®^ Human Myokine Magnetic Bead Panel instructions is <10%.

### 2.5. Dietary Intake

A prospective method of three food records was used to estimate DI during the week before the marathon race (3rd to 8th April in 2017, and 2nd to 7th April in 2018). Dietetics instructed the runners to fill in the meal time and all food and drinks that were ingested, including portion size and food brand, on two days of the week and one day of the weekend. In an interview with the runners one day before the race, the food records were checked by a trained nutrition undergraduate student to elucidate or complete missing food data. The energy intake (kcal), macronutrients (g or g/kg), and micronutrients (mg) were estimated by the professional Dietbox (http:/dietbox.me) website/app. The United States Department of Agriculture–Agricultural Research Service (USDA) food composition database and Brazilian Table of Food Composition database (TACO, University of Campinas, São Paulo, SP, Brazil) were used in the professional Dietbox website/app to provide the nutrient composition of foods.

### 2.6. Statistical Analyses

Data of general characteristics, DI, and myokines are reported as mean ± SEM of sixty endurance runners. Statistical analyses were performed using GraphPad Prism (GraphPad Prism version 9). The myokines were used as the independent variable. The normality of the data distribution was determined using the Kolmogorov–Smirnov test and the normality was rejected. Statistical analyses of myokines were evaluated using the Kruskal–Wallis test and Dunn’s test for multiple comparison. Correlations between myokines and DI (macronutrients and micronutrients) were performed by the Spearman test. Statistical significance was accepted at the level of *p* < 0.05 in all analyses. Statistical analyses of myokine levels in runners with an adequate and inadequate intake of the percentage of sucrose and carbohydrate in the energy intake (EI), with fiber, calcium, sodium, and potassium intakes evaluated using the Mann–Whitney test, and carbohydrate, cholesterol, selenium, vitamin B3, and phosphorus intakes evaluated using the Kruskal–Wallis test. Cohen’s d was calculated to estimate the effect size of the significant differences of myokine levels between runners with adequate and inadequate intake.

The number of runners with lower, adequate, higher, or very higher intake are described in Table 1.

## 3. Results

### 3.1. General Characteristics

The general characteristics of sixty runners are described as follows: age, 40.9 ± 1.0 years; body mass, 74.6 ± 1.3 kg; height, 1.73 ± 0.01 m; BMI, 24.9 ± 0.4 kg/m^2^; percentage of fat mass, 21.6 ± 0.6%; free fat mass, 58.3 ± 0.9 kg; race time, 258.7 ± 6.0 min, training experience, 6.9 ± 0.5 years; time in 10 km race, 46.7 ± 0.77 min; frequency of training, 4 (3.75–5.0) times/week; and training volume, 51.8 ± 2.7 km/week. The CPET parameters are: time of exhaustion, 11.5 ± 0.3 min; maximum speed of runners, 18.4 ± 0.3 km/h; anaerobic threshold oxygen consumption (VO_2_ AT), 33.6 ± 1.0 mL/kg/min; respiratory compensation point oxygen consumption (VO_2_ RCP), 51.6 ± 1.2 mL/kg/min; and peak oxygen consumption (VO_2_ peak), 54.4 ± 1.3 mL/kg/min.

### 3.2. Dietary Intake

The energy and macronutrient intake are summarized in Table 2. We observed an adequate protein, total fat, and sucrose intake; however, a low energy, carbohydrate, and fiber intake were observed, as well as a higher cholesterol intake.

The micronutrient daily intake is summarized in Table 3. We observed low folic acid, vitamin D, calcium, and magnesium intakes.

### 3.3. Correlation: Dietary Intake and General Characteristics

Percentage of fat mass was negatively correlated with EI, % of adequate EI, carbohydrate, protein, sucrose, vitamin B2, calcium, manganese, and phosphorus intake (Table 3). Free fat mass was negatively correlated with the percentage of carbohydrate and protein of EI (r = −31, *p* = 0.016; r = −26, *p* = 0.042), and carbohydrate intake (r = −27, *p* = 0.032). Race time was negatively correlated with protein, cholesterol, vitamin B3, selenium, magnesium, potassium, and phosphorus (Table 4).

The runners with lower carbohydrate (Cohen’s d = 1.37), phosphorus (Cohen’s d = 1.48), and calcium intakes (Cohen’s d = 1.16) had a higher percentage of fat mass (Figure 1A–C).

Runners with higher intakes of protein (>2 g/kg/day, Cohen’s d = 0.79), cholesterol (>600 mg, Cohen’s d = 0.98), selenium (>110 mg, 2-fold RDA, Cohen’s d = 0.99), vitamin B3 (>32 mg, 4-fold RDA, Cohen’s d = 0.86), phosphorus (>1400 mg, 2-fold RDA, Cohen’s d = 1.62), and potassium (>2000 mg, 100% RDA, Cohen’s d = 0.79) had lower race times (Figure 1D–I).

#### 3.3.1. Myokine Analyses

Running the marathon elevated BDNF (2-fold,), FSTL (2-fold), FGF-21 (4-fold), and IL-6 (5-fold) plasma levels (Figure 2). The IL-6 concentration was slightly reduced 72 h after the marathon (Figure 2D).

We also demonstrated a decrease in musclin and apelin after the race (Figure 2E,F); musclin, apelin, myostatin, and irisin levels 24 h after the marathon (Figure 2E–H); and musclin, myostatin, and IL-15 levels 72 h after the race (Figure 2E,G,I).

#### 3.3.2. Myokines and DI

Before the race, BDNF was positively correlated with the percentage of carbohydrate and sucrose in the EI, as well as fiber intake (r = 0.36, *p* = 0.01; r = 0.38, *p* = 0.0053; r = 0.30, *p* = 0.033, respectively), and negatively correlated with sodium (r = −0.27, *p* = 0.049). The percentage of carbohydrate also had a positive correlation with FSTL (r = 0.32, *p* = 0.025). Musclin, myostatin, IL-15, irisin, and apelin were not associated with dietary intake before the race (data not shown).

Runners with a sucrose intake above 10% of their EI and adequate levels of sodium intake (<2300 mg/day) had higher levels of BDNF (Figure 3A,B).

### 3.4. Myokines Induced by Exercise and Dietary Intake

After the race, carbohydrate intake (g/kg) was correlated with musclin and myostatin levels (r = 0.29, *p* = 0.027; r = 0.32, *p* = 0.014), and the percentage of carbohydrate of EI was correlated positively with musclin and FGF-21 levels (r = 0.26, *p* = 0.047; r = 0.33, *p* = 0.012).

Runners with adequate carbohydrate intake (>5 g/kg/day) had higher levels of myostatin 72 h after the race, with Cohen’s d = 1.46 (Figure 3C), and higher levels of musclin after the race and 24 h after the race, with Cohen’s d = 0.90 and 0.79 (Figure 3D,E).

In the recovery period, the percentage of carbohydrate of EI was correlated positively with IL-15 and BDNF (r = 0.34, *p* = 0.01; r = 0.30, *p* = 0.024), and sucrose intake had a positive correlation with BDNF and FSTL levels (r = 0.36, *p* = 0.0049; r = 0.30, *p* = 0.022).

Runners with less than 45% of carbohydrate of EI had lower levels of IL-15 (Cohen’s d = 0.59, 24 h after race) and BNDF (72 h after the race, Cohen’s d = 0.65) (Figure 3F,G), and those with less than 10% of sucrose had lower levels of BDNF (24 h after the race, Cohen’s d = 0.42) (Figure 3H).

Protein intake (g/kg or % of EI) was not correlated with the myokines determined in this study in all periods (before and after the race, and in the recovery period) (data not shown).

Cholesterol intake negatively correlated with apelin and irisin levels (r = −0.26, *p* = 0.037; r = −0.31, *p* = 0.015, respectively) after the race. Runners with a higher cholesterol intake (>600 mg/day) had lower levels of irisin (Cohen’s d = 0.71) and apelin (Cohen’s d = 0.96) compared to runners with adequate cholesterol intake after the race (Figure 4A,B). Cholesterol intake also showed a negative correlation with apelin 24 h after the race (r = −0.26, *p* = 0.44).

Fiber intake had a correlation with irisin and BDNF levels 72 h after the race (r = 0.34, *p* = 0.00067; r = 0.28, *p* = 0.028). Runners with a low fiber intake (<25 g/day) had lower levels of irisin (Cohen’s d = 0.49) and BDNF (Cohen’s d = 0.60) (Figure 4C,D).

After the race and in the recovery period, vitamin C was positively correlated with IL-15, musclin, FSTL, myostatin, IL-6, and FGF-21 (Table 5).

Thiamine (B1) was correlated with myostatin levels after the race (r = 0.31, *p* = 0.014), vitamin D was negatively correlated with IL-6 levels after the race (r = −0.26, *p* = 0.044), and musclin 72 h after the race (r = −0.26, *p* = 0.037). Manganese was correlated with musclin after the race (r = 0.30, *p* = 0.019), and selenium was negatively correlated with apelin 24 h after the race (r = −0.28, *p* = 0.030).

## 4. Discussion

Our study highlights the importance of carbohydrate, energy and vitamin C intake for the percentage of fat mass, race time, and myokine levels before (BDNF) and after the race (myostatin, musclin, irisin, apelin), and in the recovery period (BDNF, IL-15, FSTL, FGF-21, myostatin, and musclin). Chronic higher sodium and cholesterol, or low fiber consumption were demonstrated to reduce some myokine levels (BNDF, irisin, and apelin). Our results contribute to elucidating the mechanisms involved in the effects of low or higher nutrient intakes on tissue recovery and exercise adaptations.

In this study, amateur long-distance runners had low energy, carbohydrate, and fiber intakes and higher cholesterol intakes in accordance with scientific literature, which described low energy availability in elite and amateur athletes [12,18]. Relative Energy Deficiency in Sport (RED-S) has been reported in male and female elite athletes and impairs endocrine response (i.e., cortisol, insulin, IGF-1, adipokines, incretins), contributing to metabolic and immune dysfunction [19,20,21,22]. Previously, we suggested that pro-inflammatory cytokines induced by endurance exercise are higher in runners with a low energy, carbohydrate, and fiber intake, and the adequate carbohydrate intake tended to promote higher IL-10 levels in the recovery period of intense exercise [12]. RED (< 30 kcal·kg^−1^ FFM·day^−1^), accomplished by a low carbohydrate intake, also reduces the mobilization of fat stores, protein synthesis, metabolic rate, and glucose utilization, and the production of growth hormones [21,22]. Herein, we have demonstrated higher percentages of fat mass in runners with lower carbohydrate, phosphorus, and calcium intakes. There is crosstalk between adipose tissue, and calcium and phosphorus homeostasis, which involves some hormones such as leptin and parathormone [23].

We also observed higher percentages of runners with low energy (87.6%, <45 kcal/kg/FFM) and carbohydrate (76.7%, <5 g/kg/day) intakes. After the race and in the recovery period, we observed a positive correlation of myostatin and musclin with carbohydrate intake, and higher levels of these myokines with an adequate carbohydrate intake (5 to 8 g/kg/day). Myostatin acts in the process of protein degradation in muscle tissue via the activation of activin receptors (type I and II) leading to the phosphorylation and activation of SMAD-2 and SMAD-3, which forms a complex with SMAD-4, promoting the transcription of catabolic genes, and through the ubiquitin–proteosome system and autophagy activation [1,3,8], myostatin inhibition seems to increase the browning of WAT by activating the AMPK/PGC1-alpha/FNDC-5 pathway [24,25]. Activin receptors are also distributed in other tissues, and myostatin seems to impair the growth hormone (GH)/IGF1 axis in the liver [26]. The positive correlation of myostatin with carbohydrate intake does not contribute to our understanding of the role of carbohydrate intake in muscle mass repair after endurance exercise. However, it may elucidate the role of restricted caloric and low-carbohydrate diets, such as the Dietary Approaches to Stop Hypertension (DASH) diet, to improve muscle mass and cardiometabolic health [27].

Musclin activates PPAR-gamma and promotes mitochondrial biogenesis in WAT and skeletal muscle [28,29], and it also reduces glucose uptake in skeletal muscle through the inhibition of Akt/PKB and PPARγ and liver × receptor a (LXRα) activation [30]. Runners with adequate carbohydrate intake had higher musclin levels, which may contribute to improved exercise adaptations, such as the improvement of glucose homeostasis and the browning of adipose tissue.

After or in the recovery period, the percentage of carbohydrate of EI or sucrose intake was correlated positively with IL-15, FGF-21, BDNF, and FSTL levels, myokines that are responsible for muscle repair, whose functions include myogenesis (BDNF, FSTL, and IL-15), mitochondrial biogenesis (BDNF, IL-15), mytophagia (FGF-21, IL-15), autophagy (IL-15), satellite cell activation (BDNF), anti-inflammatory response (IL-15), vascular smooth muscle cell proliferation (IL-15), intramuscular fat oxidation, insulin sensitivity, and glucose intake (BDNF, FGF-21, IL-15) [1,2,3,5,8,31,32,33,34]. Many of these myokines act in several signaling pathways that will culminate in the activation of the transcription coactivator peroxisome proliferator-activated receptor-gamma coactivator (PGC)-1alpha and transcription factor PPAR-gamma, which modulate genes related to the muscle autocrine/paracrine effects cited above [25].

Moreover, these myokines promote crosstalk between skeletal muscle and other cardiometabolic tissues, which improves glucose homeostasis (IL-15, BDNF, and FSTL), enhances fuel utilization of glucose and lipids (FGF-21), and enacts a cardioprotective role (FGF-21 and BDNF) [1,5,9,33,34,35]. BDNF is a neurotrophin stimulated by metabolic stress and higher intracellular calcium levels, and it acts on myocardial contraction, decreasing the cardiomyocyte apoptosis and mitochondrial dysfunction, increasing motor neuron excitability and cardiomyocyte contraction, and improving lipid and glucose metabolism via p-AMPK and PGC-1α activation [31]. Interestingly, we also observed that runners with unhealthy behaviors, such as higher intakes of sodium or lower percentages of carbohydrate (<45%) or fiber intake, had lower BDNF levels.

Cholesterol intake was negatively associated with apelin and irisin levels, which improve carbohydrate and lipid metabolism, and have cardioprotection properties [8,32,36,37]. Runners with higher cholesterol intakes (>600 mg/day) had lower levels of irisin and apelin; moreover, runners with a lower fiber intake had lower levels of irisin. These myokines include a portion of the cell membrane protein, FNDC5, composed of 94-amino-acid residue fibronectin III (FNIII)-2 domains, and through the intracellular signaling pathway p38 and ERK1/2, they induce the browning of WAT, the upregulation of UCP1, and the improvement of glucose intake on skeletal muscle and insulin sensitivity [37]. Apelin seems to have a cardioprotective role, binding to the G-protein-coupled receptor (GPCR) and acting on the PI3K-Akt-NO signaling pathway, the reperfusion injury salvage kinase (RISK) pathway, the extracellular signal-related kinase 1/2 (ERK 1/2), protein kinase B/Akt, and eNOS [38]. Lipolysis on adipose tissue, lipid oxidation, and mitochondrial biogenesis on skeletal muscle are upregulated by apelin levels [39].

Muscle cells also require multiple protective enzyme systems involved in muscle function. In this study, we observed that after the race and in the recovery period, vitamin C was positively correlated with IL-15, musclin, FSTL, myostatin, IL-6, and FGF-21. Vitamin C has many physiological functions, including anti-inflammatory and anti-oxidative properties. However, the supplementation of vitamin C seems to be beneficial to vitamin deficiency, but leads to controversial responses in muscle mass and function, leading to the inhibition of protein synthesis pathways [40].

Inadequate dietary intake may influence some myokine levels responsible for the maintenance of muscle mass and function, and for the recovery of cardiometabolic tissues after endurance exercise, as well as enhancing adaptation to exercise. Professionals in the field of nutrition and sports medicine should emphasize the importance of adequate nutrition for performance improvement, the recovery of body tissues, and for the beneficial adaptations induced by exercise.

## Figures and Tables

**Figure 1 nutrients-15-00153-f001:**
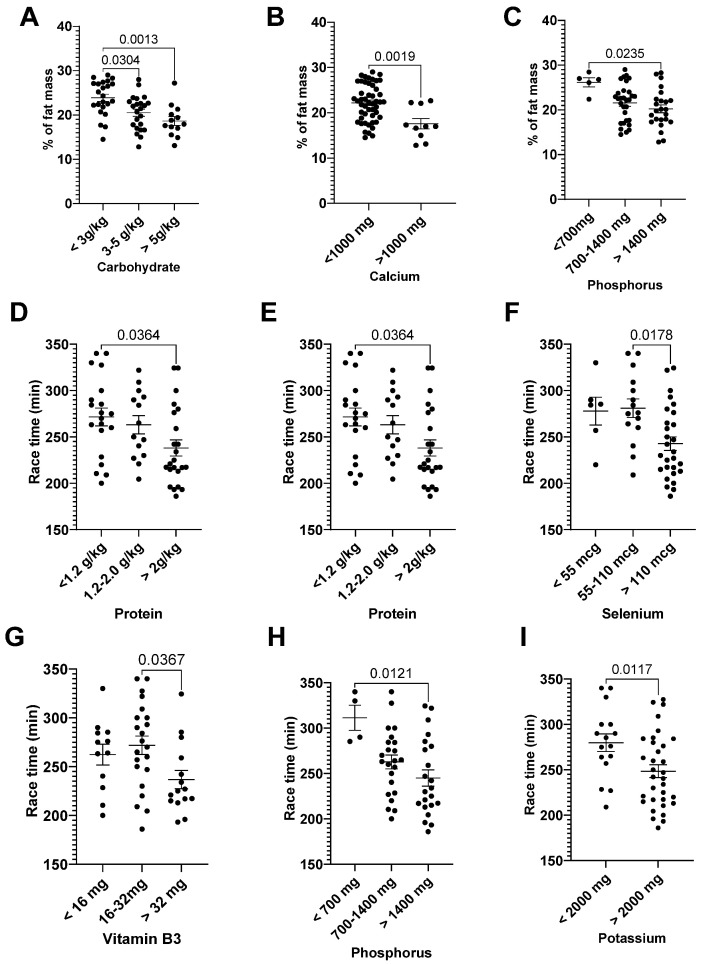
General characteristic and dietary intake. Percentages of fat mass and carbohydrate (**A**), calcium (**B**), and phosphorus (**C**) intake; race time and protein (**D**), cholesterol (**E**), selenium (**F**), vitamin B3 (**G**), phosphorus (**H**), and potassium (**I**) intake. The percentages of fat mass and race time were presented as mean ± EPM, as well as individuals’ values.

**Figure 2 nutrients-15-00153-f002:**
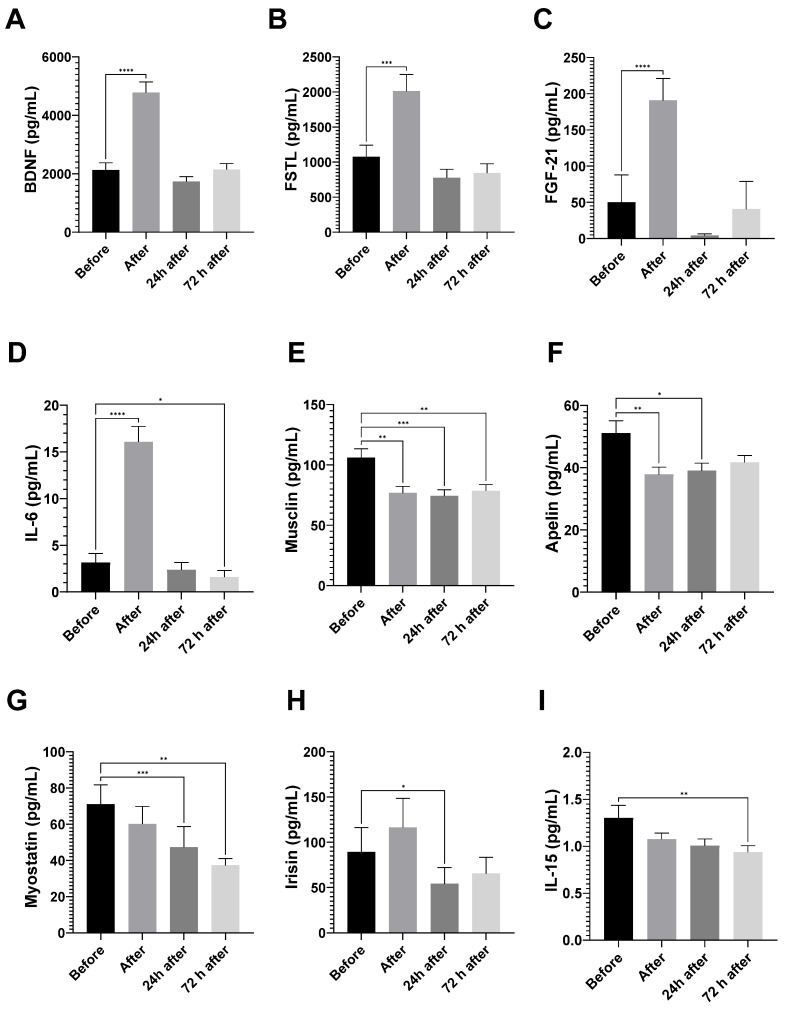
Myokine levels after the marathon race. The BDNF (**A**), FSTL (**B**), FGF-21 (**C**), IL-6 (**D**), musclin (**E**), apelin (**F**), myostatin (**G**), irisin (**H**), and IL-15 (**I**) levels are presented as mean ± EPM of 60 runners before the race, and 24 and 72 h after the race. * vs. before *p* < 0.05; ** for *p* < 0.01; *** vs. before *p* < 0.001; and **** vs. before *p* < 0.0001.

**Figure 3 nutrients-15-00153-f003:**
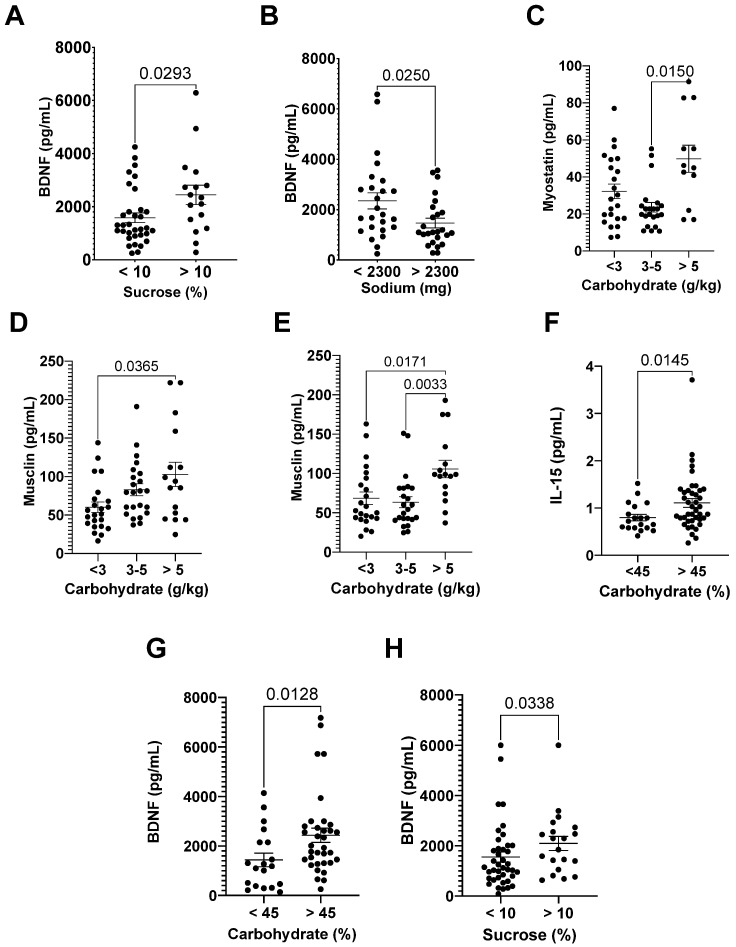
Dietary intake and myokine levels after the race and in the recovery period. The BDNF levels before the race (**A**,**B**), myostatin levels 72 h after the race (**C**), and musclin levels 24 h after (**D**) the race (**E**); IL-15 levels 24 h after the race (**F**); and BDNF levels 72 h after the race (**G**) and 24 h after the race (**H**). The myokine levels are presented as mean ± EPM, as well as individuals’ values.

**Figure 4 nutrients-15-00153-f004:**
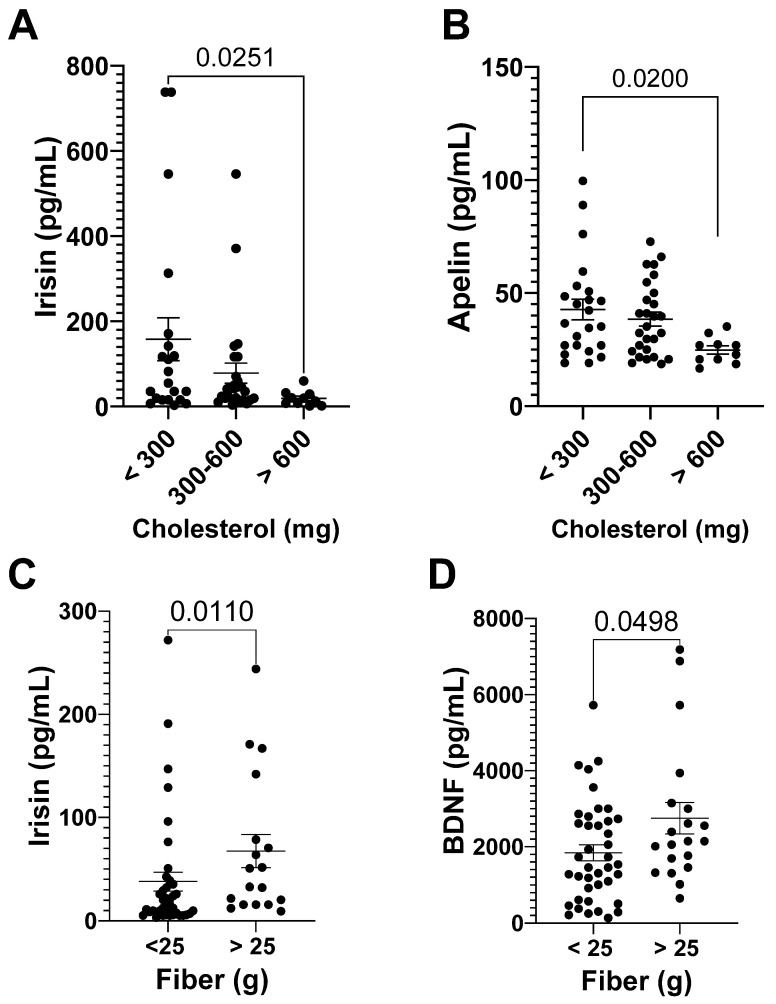
Myokine levels and cholesterol and fiber intake. Irisin (**A**) and apelin (**B**) levels after the race, and irisin (**C**) and BDNF (**D**) levels 72 h after the race. The myokine levels are presented as mean ± EPM, as well as individuals’ values.

**Table 1 nutrients-15-00153-t001:** Number of runners with lower, adequate, higher, or very higher intake.

	Lower	Adequate	Higher	Very Higher
Sucrose (% of EI)		<10	>10	
		40	20	
Carbohydrate (% of EI)	<45	45–65	>65	
	19	38	3	
Carbohydrate (g/kg/day)	<3	3–5	>5	
	23	23	14	
Protein (g/kg/day)	<1.2	1.2–2.0	>2	
	24	21	15	
Cholesterol (mg)		<300	300–600	>600
		23	27	10
Fiber (g)	<25	>25		
	40	20		
Calcium (mg)	<1000	>1000		
	50	10		
Sodium (mg)		<2300	>2300	
		29	31	
Selenium (mcg)	<55	55–110	>110	
	7	20	33	
Vitamin B3 (mg)	<16	16–32	>32	
	16	29	15	
Phosphorus (mg)	<700	700–1400	>1400	
	5	30	25	
Potassium (mg)		<2000	>2000	
		21	39	

EI, Energy Intake.

**Table 2 nutrients-15-00153-t002:** Energy, macronutrient, cholesterol, and fiber intake.

	Daily Intake	DV *
Energy intake (kcal)	2319 ± 117	2907 ± 36
Energy availability (kcal/kg of FFM)	40 ± 1.98	>45
Carbohydrate (g/kg)	3.9 ± 0.3	* 5–12
Protein (g/kg)	1.6 ± 0.1	* 1.2–1.7
Total fat (% of EI)	29 ± 1	<30%
Sucrose (% of EI)	8 ± 1	<10%
Cholesterol (mg)	391 ± 28	<300
Fiber (g)	22 ± 1	>25

* Reference daily values (DV) based on Dietary Reference Intake (DRI) of the American Dietetic Association, Dietitians of Canada, and the American College of Sports Medicine (ADA/ACSM). The values are presented as mean ± mean standard error of 60 runners.

**Table 3 nutrients-15-00153-t003:** Micronutrient daily intakes in marathon runners.

Vitamins	Daily Intake	DV *	Minerals	Daily Intake	DV *
Vitamin A (mcg)	994 ± 183	900	Calcium (mg)	715 ± 57	1000
Vitamin B1 (mg)	1.73 ± 0.14	1.2	Iron (mg)	15.7 ± 1.5	8
Vitamin B2 (mg)	1.94 ± 0.15	1.3	Mn (mg)	2.8 ± 0.30	2.3
Vitamin B3 (mg)	28 ± 3	16	Se (mcg)	163 ± 18	55
Vitamin B6 (mg)	2.3 ± 0.2	1.7	Zinc (mg)	12.7 ± 0.9	11
Folic acid (mg)	286 ± 27	400	Mg (mg)	289 ± 16	420
Vitamin B12 (mcg)	6.4 ± 1.6	2.4	P (mg)	1340 ± 73	700
Vitamin C (mg)	141 ± 28	90	Potassium (g)	2.6 ± 130	4.7
Vitamin D (mcg)	3.7 ± 0.53	15	Sodium (g)	2.5 ± 1.6	1.5
Vitamin E (mg)	12.34 ± 1.5	15			

* Reference daily values (DV) based on Dietary Reference Intake (DRI) of the American Dietetic Association, Dietitians of Canada, and the American College of Sports Medicine (ADA/ACSM). Mn, manganese; Se, selenium; Mg magnesium; P, phosphorus. The values are presented as mean ± mean standard error of 60 runners.

**Table 4 nutrients-15-00153-t004:** Correlation of dietary intake with % of fat mass and race time.

% of Fat Mass	r	*p*	Race Time (min)	r	*p*
EI (kcal/kg of FFM)	−0.28	0.025	Protein (g/kg)	−0.35	0.012
% of adequate EI	−0.34	0.007	Cholesterol (mg)	−0.40	0.043
Carbohydrate (g/kg)	−0.41	0.0009	Vitamin B3 (mg)	−0.30	0.033
Protein (g/kg)	−0.32	0.013	Se (mcg)	−0.30	0.032
Sucrose (g)	−0.34	0.007	Mg (mg)	−0.36	0.011
Vitamin B2 (mg)	−0.27	0.033	K (mg)	−0.29	0.043
Calcium (mg)	−0.49	<p.0001	P (mg)	−0.38	0.006
Mn (mg)	−0.31	0.016			
P (mg)	−0.35	0.007			

EI, energy intake; Mn, manganese; Se, selenium; Mg, magnesium; P, phosphorus. The values are presented as mean ± mean standard error of 60 runners.

**Table 5 nutrients-15-00153-t005:** Correlation of vitamin C with myokines after the race.

	*r*	*p*
IL-15 24 h after	0.33	0.010
Musclin after	0.34	0.0075
Musclin 24 h after	0.27	0.038
FSTL 24 h after	0.26	0.049
Myostatin after	0.26	0.037
Myostatin 24 h after	0.27	0.033
IL-6 24 h after	0.30	0.019
FGF-21 24 h after	0.28	0.031

IL, interleukin; FSTL, follistatin; FGF, fibroblast growth factor. Correlations between vitamin C and DDI were determined in 60 runners via the Spearman test.

## Data Availability

The data presented in this study are available in the article.

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
