# Peer review of "Chronic Low or High Nutrient Intake and Myokine Levels"

_nutrients, 2022, doi:10.3390/nu15010153_

Round 1

Reviewer 1 Report

Thank you for the opportunity to revise the manuscript entitled: "Chronic Low or High Nutrient Intake and Myokines Levels".

 The manuscript is very interesting and should review all the reference list according to the journal’s rule.

The quality of the tables is very satisfactory. The reference list exhaustively covers the relevant literature in an unbiased way. Statistical methodologies are valid and coherent with the aim of the study. All the methodology section is sufficiently documented in order to allow replication studies.

The manuscript follows a high rigor in its structure, high quality in the writing and in the quality of the content.

I believe that the manuscript will capture the interest of the audience interesting in this field.

Author Response

Thank you for the opportunity to revise the manuscript entitled: "Chronic Low or High Nutrient Intake and Myokines Levels".

Answer: We appreciate your review.

The manuscript is very interesting and should review all the reference list according to the journal’s rule.

Answer: The reference list has been revised according to the journal's guidelines.

The quality of the tables is very satisfactory. The reference list exhaustively covers the relevant literature in an unbiased way. Statistical methodologies are valid and coherent with the aim of the study. All the methodology section is sufficiently documented in order to allow replication studies.

 The manuscript follows a high rigor in its structure, high quality in the writing and in the quality of the content.

 I believe that the manuscript will capture the interest of the audience interesting in this field.

Answer: Thank you for your comments.

Miss the conclusions

Answer: Inadequate dietary intake may influence some myokine levels responsible for the maintenance of muscle mass and function, and for the recovery of cardiometabolic tissues after endurance exercise, as well as enhancing adaptation to exercise. Professionals in the field of nutrition and sports medicine should emphasize the importance of adequate nutrition for performance improvement, the recovery of body tissues, and for the beneficial adaptations induced by exercise.

The English language was reviewed by MDPI service.

Reviewer 2 Report

The study analyses the influence of inadequate intake of marathon runners on myokine levels. The manuscript has an adequate discourse and design. The results are also well represented. However, there are some doubts about the analysis that could lead to severe errors. Therefore, this should be solved. Doubts and recommendations are presented below:

Major reviews

Material and methods

Line 79: the authors should provide information on humidity, as it is a relevant data in relation to temperature for the practice of physical activity.

Line 100: 40+34 runners are mentioned at this point (74 in total), but 60 runners are used in the analysis (e.g. lines 140 and 163). Why were some runners measured in both years? If so, which measurement of those runners did you choose, 2017 or 2018? Why?

Statistical analysis

This section does not describe the variables used as independent and dependent variables.

The authors do not apply the normality or homoscedasticity test. Therefore, it is not known whether the statistical tests applied are adequate. Furthermore, the authors first use parametric analysis and then non-parametric correlations. This must be resolved or the analysis may contain severe analytical errors.

Results

This section needs to know the value of the result of each analysis, the p-value of each result and the eta squared so that readers can know the statistical power of the study.

Minor revisions

Line 146: EI is the first time it is used, so the content of the abbreviation should be specified.

Line 167: Wrong centrality value, it is not possible to train 4.3 sessions per week, because it is not a continuous variable. In this case, median and interquartile range should be used to describe the variable properly.

Author Response

Reviewer 2

Major reviews

Material and methods

Line 79: the authors should provide information on humidity, as it is a relevant data in relation to temperature for the practice of physical activity.

Answer: Thank you for your comments and suggestion. We included the average relative humidity: Lines 79-80: (average temperature 19.8 °C, average relative humidity 72.8%, 2017 and 19.9 °C, average relative humidity 87.7%, 2018)

Line 100: 40+34 runners are mentioned at this point (74 in total), but 60 runners are used in the analysis (e.g. lines 140 and 163). Why were some runners measured in both years? If so, which measurement of those runners did you choose, 2017 or 2018? Why?

Answer: Myokine determination was performed only once in each runner, 40 runners in 2017 and 34 runners in 2018.  However, from seventy-four marathon runners, sixty filled three food records in the week before marathon race for DI analyses. So, we excluded fourteen runners to the association between myokines and DI. (Lines 74 to 76).

Statistical analysis

This section does not describe the variables used as independent and dependent variables.

Answer: The myokines and dietary intake are used as independent variable.

The authors do not apply the normality or homoscedasticity test. Therefore, it is not known whether the statistical tests applied are adequate. Furthermore, the authors first use parametric analysis and then non-parametric correlations. This must be resolved or the analysis may contain severe analytical errors.

Answer: Thank you for your correction. The statistical description was wrong and we apology for this mistake. Statistical analyzes of myokines were evaluated using Kruskal Wallis test and Dunn´s test for multiple comparison. The normality of the data distribution was determined using the Kolmogorov-Smirnov test and the normality was rejected. We correct the statistical description.

Lines 141 to 144:  The myokines were used as independent variable. The normality of the data distribution was determined using the Kolmogorov–Smirnov test and the normality was rejected. Statistical analyses of myokines were evaluated using Kruskal–Wallis test and Dunn´s test for multiple comparison. Correlations between myokines and DI (macronutrients and micronutrients) were performed by Spearman test. Statistical significance was accepted at the level of p<0.05 in all analyses. Statistical analyses of myokines levels in runners with adequate and inadequate intake of percentage of sucrose and carbohydrate in the energy intake (EI), fiber, calcium, sodium and potassium intakes were evaluated using Mann-Whitney test, and of carbohydrate, cholesterol, selenium, vitamin B3 and phosphorus intakes were evaluated using Kruskal-Wallis test. Cohen´s d was calculated to estimate effect size of significantly differences of myokines levels between runners with adequate and inadequate intake.

Results

This section needs to know the value of the result of each analysis, the p-value of each result and the eta squared so that readers can know the statistical power of the study.

Answer: We included p value of each results in the text or in the graph. Cohen´s d was calculated to estimate effect size of significantly differences of myokines levels between runners with adequate and inadequate intake instead eta squared (suggested by ANOVA analyses).

Minor revisions

Line 146: EI is the first time it is used, so the content of the abbreviation should be specified.

Answer: Thank you for your correction (line 149).

Line 167: Wrong centrality value, it is not possible to train 4.3 sessions per week, because it is not a continuous variable. In this case, median and interquartile range should be used to describe the variable properly.

Answer: We included the median and Q1 and Q3 values (Line 165).

Reviewer 3 Report

line 139 to 160 must be presented in a table.

English must be reviewed. 

Miss the conclusions

Author Response

Reviewer 3

line 139 to 160 must be presented in a table.

Answer: Thank you for your suggestion. We included a Table.

The number of runners with lower, adequate, higher intake or very higher are described in Table 1.

Table 1. Number of runners with lower, adequate, higher or very higher intake.

Lower

Adequate

Higher

Very higher

Sucrose

(% of EI)

<10

>10

40

20

Carbohydrate

(% of EI)

<45

45-65

>65

19

38

3

Carbohydrate (g/kg/day)

<3

3-5

> 5

23

23

14

Protein (g/kg/day)

<1.2

1.2-2.0

>2

24

21

15

Cholesterol (mg)

<300

300-600

>600

23

27

10

Fiber (g)

<25

>25

40

20

Calcium (mg)

<1000

>1000

50

10

Sodium (mg)

<2300

>2300

29

31

Selenium (mcg

<55

55-110

>110

7

20

33

Vitamin B3 (mg)

<16

16-32

>32

16

29

15

Phosphorus (mg)

<700

700-1400

>1400

5

30

25

Potassium (mg)

<2000

>2000

21

39

English must be reviewed. 

Answer: The English language was reviewed by MDPI service.

Round 2

Reviewer 2 Report

All questions and recommendations were adequately answered and resolved by the authors.